# Patterns of Variation and Chemosystematic Significance of Phenolic Compounds in the Genus *Cyclopia* (Fabaceae, Podalyrieae)

**DOI:** 10.3390/molecules24132352

**Published:** 2019-06-26

**Authors:** Maria. A. Stander, Herman Redelinghuys, Keabetswe Masike, Helen Long, Ben-Erik Van Wyk

**Affiliations:** 1Department of Biochemistry, University of Stellenbosch, Private Bag X1, Matieland 7600, South Africa; kmasike@sun.ac.za; 2Mass Spectrometry Unit, Central Analytical Facility, University of Stellenbosch, Private Bag X1, Matieland 7600, South Africa; 3CREST (Centre for Research on Evaluation, Science and Technology), University of Stellenbosch, Private Bag X1, Matieland 7600, South Africa; hredelinghuys@sun.ac.za; 4Department of Botany and Plant Biotechnology, University of Johannesburg, P.O. Box 524, Auckland Park, Johannesburg 2006, South Africa; phytomed@uj.ac.za (H.L.); bevanwyk@uj.ac.za (B.-E.V.W.)

**Keywords:** *Cyclopia*, honeybush tea, phenolic compounds, orobol, butein, mangiferin, LCMS, isosakuranetin, multivariate data analysis

## Abstract

As a contribution towards a better understanding of phenolic variation in the genus *Cyclopia* (honeybush tea), a collection of 82 samples from 15 of the 23 known species was analysed using liquid-chromatography–high resolution mass spectrometry (UPLC-HRMS) in electrospray ionization (ESI) negative mode. Mangiferin and isomangiferin were found to be the main compounds detected in most samples, with the exception of *C. bowiena* and *C. buxifolia* where none of these compounds were detected. These xanthones were found to be absent from the seeds and also illustrated consistent differences between species and provenances. Results for contemporary samples agreed closely with those based on analysis of a collection of ca. 30-year-old samples. The use of multivariate tools allowed for graphical visualizations of the patterns of variation as well as the levels of the main phenolic compounds. Exclusion of mangiferin and citric acid from the data was found to give better visual separation between species. The use of UPLC-HRMS generated a large dataset that allowed for comparisons between species, provenances and plant parts (leaves, pods, flowers and seeds). Phenetic analyses resulted in groupings of samples that were partly congruent with species but not with morphological groupings within the genus. Although different provenances of the same species were sometimes found to be very variable, Principle Component Analysis (PCA) indicated that a combination of compounds have some (albeit limited) potential as diagnostic characters at species level. 74 Phenolic compounds are presented, many of which were identified for the first time in *Cyclopia* species, with nine of these being responsible for the separation between samples in the PCAs.

## 1. Introduction

*Cyclopia* Vent. is a fynbos-endemic genus of legumes (family Fabaceae, tribe Podalyrieae) comprising 23 known species. Several species have a long history of traditional use as herbal teas [1] but it is only recently that commercial crop and product development has been initiated [2,3], focused mainly on *C. genistoides* (L.) R.Br., *C. intermedia* E.Mey. and *C. subternata* Vogel. These three species are generally referred to as *heuningbostee*, *bergtee* and *vleitee*, respectively. Other species such as *C. sessilifolia* Eckl. & Zeyh. (*Heidelbergtee*) and *C. maculata* (Andrews) Kies (*Genadendaltee*) have also been used to a limited extent [4,5,6,7,8]. The species are superficially rather similar, resulting in a confused taxonomy and nomenclature [4,5,6,7]. Infrageneric relationships are complicated by the fire-survival strategies of the species because the distinction between seeding and sprouting is not always clear, and some overlap seems to occur [6]. Based on extensive field studies in the early 1990’s, a detailed revision of the genus was published, in which the delimitation and geographical distribution of the species were clarified [7]. As part of a broader chemosystematic study of Cape genistoid legumes, reviewed in 2003 [9], an attempt was made to compare *Cyclopia* with other genera of the tribe Podalyrieae. It was found that *Cyclopia* species do not accumulate quinolizidine alkaloids as is typical for other genera, but that the leaves were rich in phenolic compounds. *Cyclopia* proved to be chemically distinct from other genera of the tribe, indicating an isolated phylogenetic position [10,11,12,13]. De Nysschen and co-workers [14] were the first to isolate and describe mangiferin as the main phenolic compound, which co-occurred with hesperitin and isosakuretin in leaves. Another study [15] showed that butein, 3’hydroxydaidzein and other flavonoids are the main seed metabolites, not only in *Cyclopia* but also in other genera of the tribe. The HPLC system used at that time [14], [15] did not provide sufficiently accurate quantitative data to distinguish between the species. 

Liquid-chromatography–high resolution mass spectrometry (UPLC-HRMS) has previously been used for the analysis of *C. subternata* [16,17,18] and *C. genistoides* [19], but as yet, no studies have been conducted into the full extent of chemical variation in the rest of the genus (including the non-commercial species). It was decided to reinvestigate *Cyclopia* species with the aim of not only describing the phenolic variation in the genus (which is relevant to developing better quality control analyses) but also to have another attempt at evaluating the chemosystematic significance of the main phenolic compounds. Several authentic samples used by Schutte [7] in her revision of the genus were available for study. The aim was to determine if different species and populations of *Cyclopia* could be distinguished by quantitative and perhaps also qualitative differences in their overall phenolic profiles.

## 2. Results and Discussion

Table 1 lists the main compounds detected, while Table 2 contains a list of the samples, their species, voucher numbers and collection localities. Figure 1 shows the total ion chromatograms of four different *Cyclopia* species and highlights the differences in phenolic profiles that were detected. The tentative identification of compounds was based on previous papers [16,17,18,19,20,21,22,23,24,25,26], as well as a combination of fragmentation data, elemental composition based on accurate mass, relative retention times and UV data.

### 2.1. Tentative Identification of New Compounds in Cyclopia in Table 1

Two isomeric peaks with *m/z* 429 [M − H]^−^, (C_19_H_25_O_11_), from compounds **12** and **13,** eluted at retention times (Rts) of 11.21 and 11.39 min respectively. The MS^E^ spectra at higher collision energy (function 2) showed an intense fragment ion (base peak or bp) at *m/z* 135 (C_8_H_7_O_2_) for both peaks. The molecular formula for this fragment ion corresponds to that of piceol (4-hydroxyacetophenone), previously identified in the methanolic extracts of *Cyclopia genistoides* [20]. Metabolites 12 and 13 were thus tentatively annotated as piceol-*O*-hexose-*O*-pentoside isomers, with the piceol fragment ion being produced by neutral loss of a disaccharide moiety (−294 Da) consisting of hexose (−162 Da) and pentose (−132 Da) subunits.

At Rt 11.68 min, a peak with a *m/z* of 443 [M − H]^−^, (C_20_H_27_O_11_), compound **16**, was observed, with corresponding fragment ions at *m/z* 135 and 96. It is not clear what the ion at *m/z* 96 represents, but the calculated molecular formula of the *m/z* 135 fragment corresponds to that of piceol (as discussed above), arising from the neutral loss of a disaccharide moiety (−308 Da) with hexose and rhamnoside (−146 Da) subunits. Thus, this peak was tentatively annotated as piceol-*O*-hexose-*O*-rhamnoside.

Compound **51**, eluting at Rt 20.17 min presented a precursor ion at *m/z* 445 [M − H]^−^, (C_22_H_21_O_10_), which under MS^E^ fragmentation showed fragment ions at *m/z* 283 [M − H]^−^, (C_16_H_11_O_5_), 286 and 239. The molecular formula of the aglycone fragment at *m/z* 283 corresponds to the isoflavone olmelin (biochanin A) [22], produced by neutral loss of a hexose moiety, whilst the ion at *m/z* 268 results from the further neutral loss of the methyl (−15 Da) group. Thus, this peak was tentatively annotated as olmelin (biochanin A)-*O*-hexoside.

At Rt 21.16 min, a peak with *m/z* 271 [M − H]^−^, (C_15_H_11_O_5_), compound **56,** was observed, with fragment ions of *m/z* 135 and 91 which are characteristic of both the flavanone butin and the chalcone butein [23]. Compound 56 was thus tentatively annotated as butin/butein. Two isomeric peaks eluting at Rt 22.39 and 23.49 min with a *m/z* 433 [M − H]^-^, (C_21_H_21_O_10_) were observed. The MS^E^ spectra showed fragment ions similar to those observed for butin/butein, namely 271, 135 and 91. Since the fragment ion at *m/z* 271 results from the neutral loss of a hexose moiety, these peaks were tentatively annotated as butin/butein-*O*-hexoside isomers.

### 2.2. Levels of Main Compounds

Figure 2 consists of a heatmap of the main compounds detected in the samples, showing the higher concentration compounds in lighter shades and low concentrations in dark, note the many light blocks for citric acid and mangiferin. Since calibration standards are not available for the majority of compounds detected, the peak areas for these compounds were converted to concentration values in mg/kg by interpolation off the mangiferin calibration curve and are provided in the Appendix A. Mangiferin levels in the plant extracts were found to be above the linear range of the mass spectrometer and their concentrations in this table should therefore be seen as relative. The mangiferin levels of 30 of the samples were more accurately determined using UV detection at 280 nm. Concentrations of between 0.41 and 3.8 g/100 g were recorded in the samples where the compound was present (Results not shown).

The phenolic metabolites of *Cyclopia* species that have been commercialized (*C. subternata, C. genistoides, C. intermedia*) have been well studied and the results thereof published extensively [16,17,18,19,20,24]. In addition, some mention is also made of *C. sessiliflora* and *C. maculata* which are also commercially processed, albeit on a smaller scale [18,25]. Walters et al. [21] investigated the phenolic composition of the non-utilised species *C. pubescens* and detected the xanthones mangiferin and isomangiferin as some of the main compounds. The same authors also detected flavanones, a flavone and benzophenones. Methylated flavonoids including the isoflavone, formononetin and afrormozin as reported by [20] in *C. subternata* were not reported by other investigators. The reported presence [14] of (iso)sakuranetin and hesperitin, which elute rather late in the chromatogram was confirmed in this study (Table 1). It is possible that these compounds may not elute off the C_18_ column of a modern reverse-phase chromatographic system, since the work of earlier investigators was performed on normal phase systems. This scenario was investigated by extracting one sample with solvents of different polarity (methanol, dichloromethane, dimethylsulfoxide, ethanol, water and combinations of these). The analysis was then repeated using the current method as well as on a much shorter column using a stronger gradient. The results showed a lower extraction efficiency of early eluting polar molecules and a higher extraction efficiency for non-polar late eluting molecules when using stronger solvents. For example, 20% more luteolin and 33% less mangiferin was extracted using methanol/dichloromethane compared to 50% methanol. No other methoxylated flavonoids were detected using this solvent system, only some hydroxylated long chain fatty acids were detected (Figure 3).

Figure 4 contains the structures of selected compounds presented in Table 1. The PCA cluster map of all the samples is presented in Figure 5. Two *Cyclopia* species that do not produce mangiferin (*C. buxifolia*, BX and *C. bowieana*, BW) are seen as outliers on the right hand side. The clustering was driven by mangiferin and the rest of the species were not visually well separated in the cluster map. In addition, the samples from flower parts other than the leaves (twigs, stems and flowers), also influenced the separation. Figure 6 is the cluster map of only the leaf samples with the mangiferin data excluded. The groupings of the species in clusters improved somewhat with e.g. *C. genistoides* now clustering on its own. In Figure 7 only the leaf extracts of the three commercial species *C. intermedia* (IN), *C. genistoides* (GE), and *C. subternata* (SU) were investigated with citric acid and mangiferin excluded. This showed a separation of *C. genistoides* (green, cluster 2,5, and 6) from *C. subternata* (orange, cluster 1) and *C. intermedia* (red, cluster 3,4), with some extracts forming additional clusters that appear to be based on geography/provenance/population. 

### 2.3. Old Samples Versus Contemporary Samples

No significant differences between older and newer sample were detected which confirms the stability of these phenolic compounds in plants if stored as dry material.

### 2.4. Differences Between Plant Parts (Twigs, Leaves, Pods, Flowers And Seeds)

This study has shown that the same compounds occur at varying concentrations in different plant parts, with the exception of the seeds that contain certain unique compounds but lack others, especially the flavonoid glycosides (Figure 8). A comparison of the main classes of compounds between plants parts is presented in Figure 9. There are only quantitative differences between twigs, leaves and pods in *Cyclopia aurescens* Kies, but the seeds are markedly different, with a dominance of chalcones and flavanones. The major seed flavonoids in *Cyclopia* were reported by De Nysschen et al. [15] as butin, 3’-hydroxydaidzein, butein and vicenin-2, but these compounds have not been detected in more recent studies. In our study, butein/butin and derivatives were detected in seeds at much higher levels than in the leaves, pods or stems. We have recorded a significant peak for 3’hydroxydaidzein (one of the main compounds detected in seeds by De Nysschen [14,15] in one of the seed samples (AU5S, *Cyclopia aurescens* Kies). This peak corresponds to 3’hydroxydaidzein (*m/z* 269.0451, C_15_H_9_O_5_ fragment ions: 269.0453 (base peak), 133.0294, retention time 20.9, eluting just before the butein peak in Figure 8). Vicenin-2 is also more prominent in the seed samples, but co-elutes with isomangiferin in the extracts from twigs, leaves, pods and flowers.

### 2.5. Diagnostic Value of Phenolic Compounds

The results suggest that phenolic compounds do have diagnostic value in distinguishing between some of the species, especially when combinations of some of the compounds are used. Figure 10 shows the average composition of compounds for the species studied. *Cyclopia buxifolia* and *C. bowieana* are apparently unique in their inability to produce xanthones and benzophenones; this chemical difference presumably makes them unsuitable for tea production. The other species have similar combinations of compounds, but the relatively high levels of xanthones in *C. genistoides* must be noted. The seemingly random quantitative combinations of main compounds in leaf samples of all the species are shown in Figure 11 comparing the concentrations of the individual flavanones. There is visually no clear pattern in Figure 11 and the underlying processes (phenotypic or genetic) deserve more detailed studies. A somewhat clearer picture emerges when multiple samples from different provenances are analysed, as shown in Figure 12 that represents flavanones of the commercial species: *C. genistoides*, *C. intermedia* and *C. subternata*. Note that different plants collected from the same population often have very similar chemical profiles, while different populations tend to be somewhat different. From this result it is clear that a large part of the chemical variation in the three commercial species can be ascribed to provenance. Chemical differences at population level are often genetically determined and it will be interesting to compare cultivated plants with plants from the original populations where the seeds were collected. A similar pattern emerges when the phenolic compounds from the loading plots that caused the separation of clusters in Figure 7 are considered (Figure 13). Note that the unique combinations of compounds that are uniform within a provenance are often discontinuous between all or most of the species. The chemical identities and the diagnostic value of the nine compounds shown in Figure 13 should be a priority for future studies. This would require isolation and purifying the compounds and confirmation and structural elucidation using Nuclear Magnetic Resonance spectroscopy (NMR).

When mangiferin and citric acid were removed from the data set, distinct clusters were obtained. Cluster analysis, however, often grouped extracts from the same species together but many were not congruent with species delimitations, i.e. clustering was based on provenance rather than species (see Appendix A). The dendrogram also did not group species together that are presumed to be related on the basis of morphological characters. *Cyclopia genistoides* differs from *C. subternata* and the majority of provenances of *C. intermedia* in the higher concentrations of mangiferin. *Cyclopia intermedia* is a widely distributed species with some morphological differences between populations and it seems that some outlier values may obscure what is otherwise a promising diagnostic difference. Stepanova et al. [24] found leaf anatomical characters to distinguish between *C. genistoides*, *C. intermedia* and *C. subternata* but chemical analyses are clearly a more practical approach for quality control purposes. Particular provenances are usually selected for crop development, so that commercial tea samples are likely to be chemically more uniform than wild-harvested material collected from unknown populations. Developers often try to standardise the chemical composition of herbal products in order to minimize batch to batch variation. In this context, the numerous chemical compounds and their diversity in *Cyclopia* species described here are likely to provide a practical and reproducible approach to identify the source species of the material, to detect possible contaminants and assess the quality of the product. 

## 3. Conclusions

The analyses of *Cyclopia* species using UPLC-HRMS with simultaneous collection of low collision energy MS data, ramped collision energy MS data and UV data resulted in large, complex datasets, which revealed considerable complexity in the phenolic compounds observed. MS^E^ fragmentation data is presented for 74 phenolic compounds, including at least three benzophenones, two dihydrochalcones, three chalcones, three xanthones, 17 flavanones, three flavones, two isoflavones, three acetophenones and eight phenolic acids (cinnamic acid derivatives). Some unknown compounds have been tentatively identified including piceol-hexose-pentoside isomers, piceol-hexose-rhamnoside, butein-hexosides and olmelin-*O*-hexoside.

The study also revealed that the methods of extraction and analysis by UPLC-HRMS analysis influence the results and that both polar and nonpolar (methylated) compounds may be overlooked in routine analyses. Plant parts (twigs, leaves, flowers and pods) show only quantitative differences in the main constituents but seeds often contain much lower concentrations of xanthones and higher concentrations of chalcones and other flavonoids. As suggested in the literature, phenolic compounds have limited chemosystematic value at species level but a combination of chemical characters can be used to distinguish between some of the species. The study provides deeper insights into the chemical complexity of *Cyclopia* species and the potential role that UPLC-HRMS analyses can play, not only in quality control but also to help select superior chemotypes for crop and product development.

## 4. Materials and Methods

Methods and equipment were the same as used by Stander et al., [25] but the gradient was extended to 37 minutes to accommodate more non-polar compound including isoflavones and methoxylated flavonoids described in previous papers [14,17].

### 4.1. Samples and Sampling

The samples came from a collection of what are now historical materials that formed part of a comprehensive revision of the genus *Cyclopia* by Schutte [7], who also identified the materials (Table 2). De Nysschen [14] used part of this collection for a study of the main phenolic compounds in the genus, and reported the presence of mangiferin as the main constituent for the first time. The material was carefully stored at low humidity in a dark storeroom. We have previously shown [25] that the main phenolic compounds of commercial rooibos tea are remarkable stable, producing almost identical phenolic profiles after more than 80 years of storage. 

### 4.2. Extraction 

Depending on available material, ca. 300 to 500 mg of dry plant material was soaked overnight in 50% methanol in water containing 1% formic acid (2 mL), using 15 mL polypropylene centrifuge tubes. The volumes of solvent were adjusted according to the available sample amount to 7.5 mL per 1 gram of sample. The samples were extracted in an ultrasonic bath (0.5 Hz, Integral systems, RSA) for 60 min at room temperature, followed by centrifugation for 5 minutes (Hermle Z160m, 3000× *g*) and transferred to glass vials.

### 4.3. Standards

Standards were obtained from Sigma-Aldrich: mangiferin, citric acid, naringenin, hesperidin, kaempferol, quercetin and ferulic acid were analytically weighed out and dissolved in dimethyl sulfoxide (DMSO) and diluted in methanol to a calibration series of 2, 5, 10, 40, 50, 100, 200, 500 mg/L.

### 4.4. UPLC-HRMS Analysis

UPLC-HRMS analysis was performed using a Waters Synapt G2 Quadrupole time-of-flight (QTOF) mass spectrometer (MS) connected to a Waters Acquity ultra performance liquid chromatograph (UPLC) (Waters, Milford, MA, USA) with photodiode array detector. A Waters HSS T3, 2.1 × 150 mm, 1.7 µm column with water with 0.1% formic acid in line A and 0.1% formic acid in acetonitrile in line B. A flow rate of 0.25 mL/min was used and the gradient started with 100% solvent A for 1 minute followed by a linear gradient to 28% B in 21 minutes and another linear gradient to 60% B in 8 minutes. The column was washed for 1 minute at 100% B and then re-equilibrated.

Data were acquired in MS^E^ mode whereby a low collision energy scan is followed by a high collision energy scan to obtain both molecular ion [M − H] and fragment data at the same time. During the high collision energy scan the collision energy was ramped from 20 to 60V. Electrospray ionisation was used in the negative mode and a scan range of 120 to 1500 was used. The desolvation temperature was set at 275 °C and nitrogen was used as desolvation gas at 650 L/h. The capillary voltage was 25 V and the instrument was calibrated with sodium formate and leucine encephalin was used as lock mass for accurate mass determinations.

### 4.5. Data Processing and Clustering

The Markerlynx application manager of MassLynx™ version 4.1 software (Waters Corporation, Boston) was used to align the raw mass spectrometry data and convert it to retention time-mass pairs with signal intensity for each peak. Selected mass peaks from the mass spectra were normalised to compensate for the variance in concentration and ensure equal representation in the dataset, thereby facilitating comparative analysis. Normalisation involves scaling each sample vector using least squares normalisation (L2 norm), independently of other samples. Multivariate analysis was performed similar to [25].

Principal component analysis (PCA) was performed on the dataset. The number of PCA components was selected so that the amount of variance that needs to be explained is greater than two times standard deviation (95.45%) data coverage. In traditional methods, the PCA components are visualised in pairs while the loadings plot for all PCA components is displayed simultaneously. However, all the selected PCA components need to be considered collectively for meaningful discrimination of the dataset. To achieve this, unsupervised hierarchical clustering analysis was then performed on the selected PCA components. An implementation of the Mean Shift clustering algorithm was chosen as it holds no intrinsic hypothesis about the number of clusters, nor the shape thereof. This is in contrast with to the classic K-means clustering approach where the number of clusters is predetermined. Mean Shift is a non-parametric centroid based algorithm, using a radial basis function (RBF) kernel, where each point in the feature space corresponds to the initial centroid positions. It iteratively updates centroids to be the mean of all the points within a given region, thereby discovering dense regions in the feature space, until convergence was achieved. The remaining set of centroids after convergence, being the cluster centres and the data points associated with the same centroid, are members of the same cluster.

Next, the loadings factors for each PCA component was analysed, to gain an understanding of which metabolites contributed to the most variation within the dataset. The loadings plots of the Markerlynx data as well as a manual peak picking process was used to identify the main compounds in the samples. The Targetlynx application manager was then used to create a smaller subset of 74 compounds that was processed in the same way, yielding similar results. The Targetlynx dataset is reported, as it contains data with tentatively identified compounds.

## Figures and Tables

**Figure 1 molecules-24-02352-f001:**
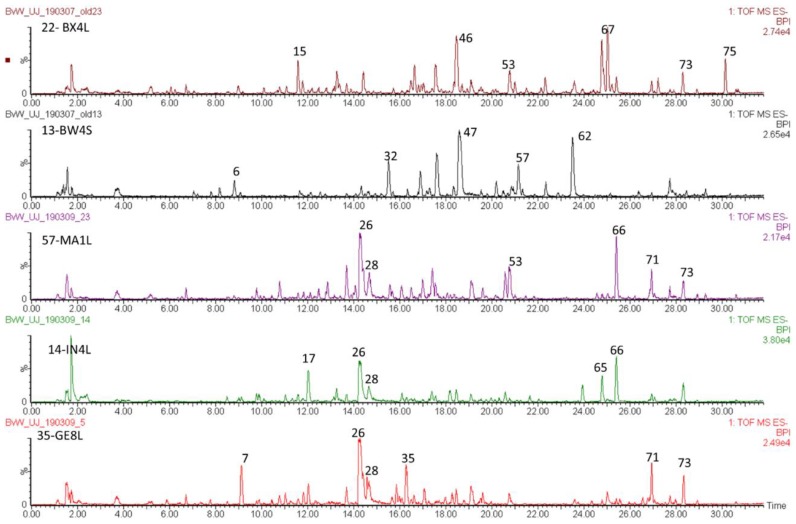
Total ion chromatograms of extracts of (from top down): *Cyclopia buxifolia* (Burm.f.) Kies leaves from Jonkershoek (BX4L); *Cyclopia bowieana* Harv. stems from Ruitersberg (BW4S); *Cyclopia maculata* (Andrews) Kies leaves from Garcia State Forest (MA1L); *Cyclopia intermedia* E.Mey. leaves from Oudtshoorn (IN4L); *Cyclopia genistoides* (L.) R.Br. leaves from Bettys Bay (GE8L) showing large differences in their phenolic profiles with mangiferin (compound **26**) absent in the top two plant extracts.

**Figure 2 molecules-24-02352-f002:**
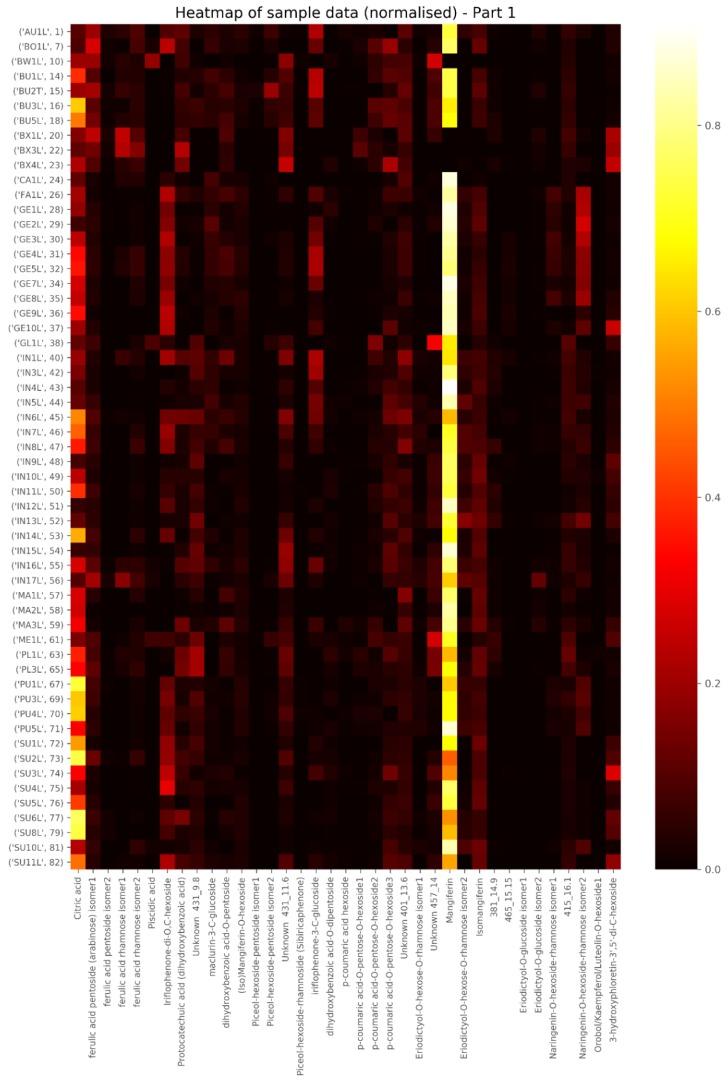
Heatmap of the main peaks detected in the *Cyclopia* extracts, showing mangiferin as the most abundant phenolic compound in most samples (light vertical line). The lighter the spot, the higher the concentration. *C. aurescens* (AU), *C. bolusii* (BO), *C. bowiena* (BW), *C. burtonii* (BU), *C. buxifolia* (BX), *C. capensis* (CA), *C. falcata* (FA), *C. genistoides* (GE), *C. glabra* (GL), *C. intermedia* (IN), *C. maculata* (MA), *C. meyeriana* (ME), *C. plicata* (PL), *C. pubescens* (PU) and *C. subternata* (SU); (L = leaves, T = twigs, P = pods, S = seeds); Sample numbering is according to sample locality from West to East per species.

**Figure 3 molecules-24-02352-f003:**
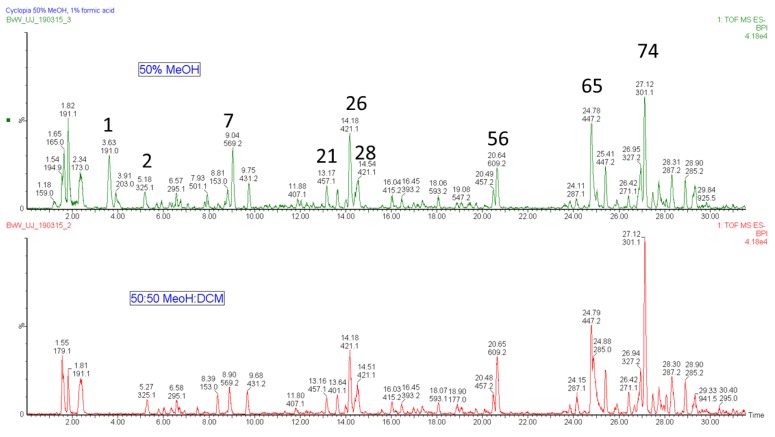
Total ion chromatogram of *Cyclopia subternata* extract of methanol/dichloromethane (1:1, **bottom**) and 50% methanol, 1% formic acid (**top**) showing different extraction efficiencies according to the polarity of the analytes.

**Figure 4 molecules-24-02352-f004:**
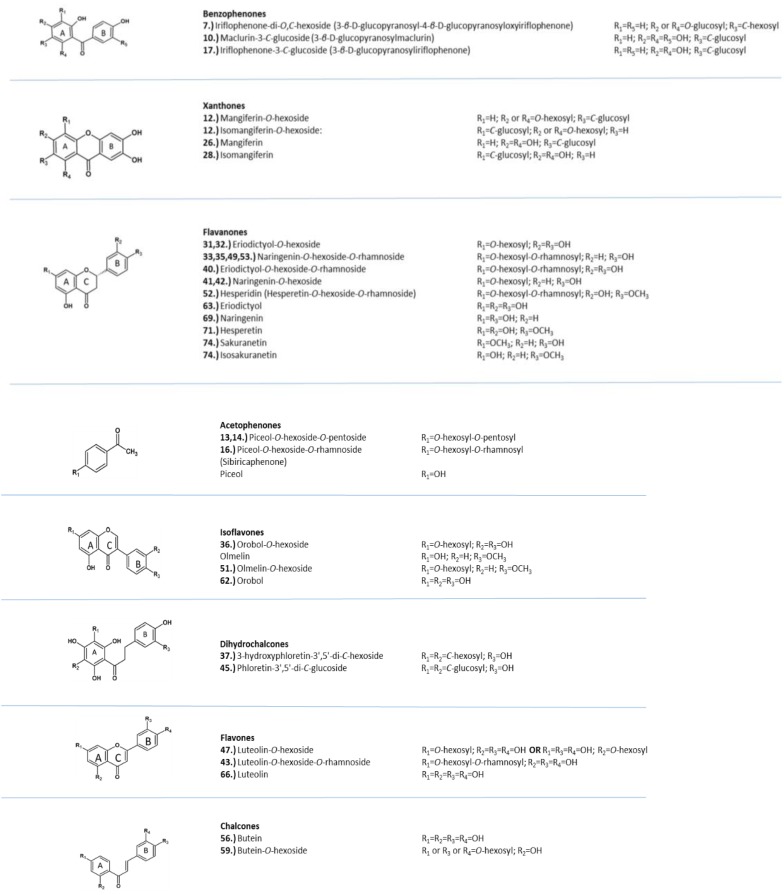
Selected structures of the compounds detected in *Cyclopia* extracts, numbered according to Table 1.

**Figure 5 molecules-24-02352-f005:**
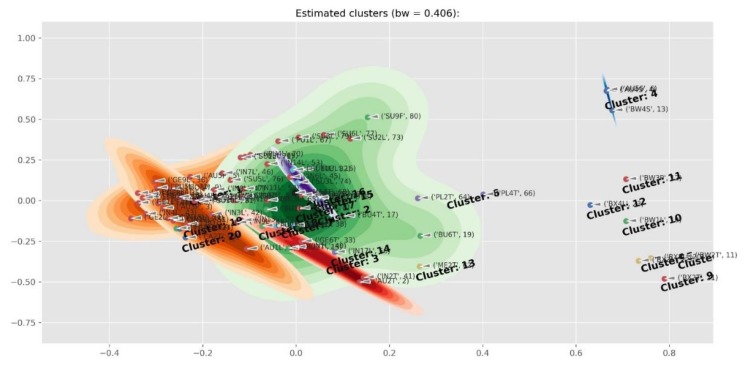
Cluster map showing the two *Cyclopia* species that apparently do not produce mangiferin (*C. buxifolia*, BX and *C. bowieana*, BW) as outliers on the right hand side. *C. aurescens* (AU), *C. bolusii* (BO), *C. bowiena* (BW), *C. burtonii* (BU), *C. buxifolia* (BX), *C. capensis* (CA), *C. falcata* (FA), *C. genistoides* (GE), *C. glabra* (GL), *C. intermedia* (IN), *C. maculata* (MA), *C. meyeriana* (ME), *C. plicata* (PL), *C. pubescens* (PU) and *C. subternata* (SU).

**Figure 6 molecules-24-02352-f006:**
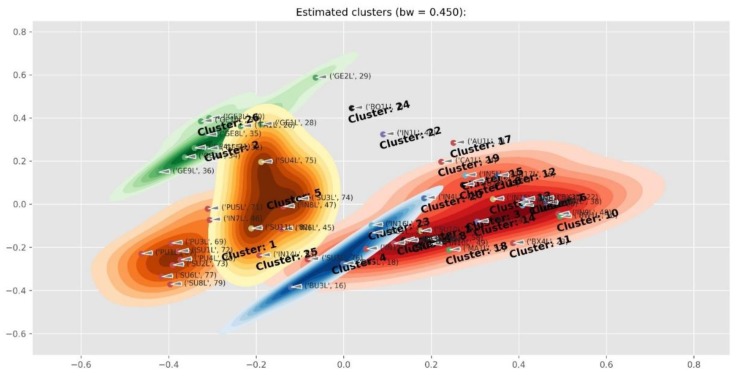
Cluster map of only the leaf samples of *Cyclopia* species with mangiferin excluded, showing an improved separation of the main cluster in Figure 5. Note, for example, that the *C. genistoides* samples now form a cluster 2 (shown in green). *C. aurescens* (AU), *C. bolusii* (BO), *C. bowiena* (BW), *C. burtonii* (BU), *C. buxifolia* (BX), *C. capensis* (CA), *C. falcata* (FA), *C. genistoides* (GE), *C. glabra* (GL), *C. intermedia* (IN), *C. maculata* (MA), *C. meyeriana* (ME), *C. plicata* (PL), *C. pubescens* (PU) and *C. subternata* (SU).

**Figure 7 molecules-24-02352-f007:**
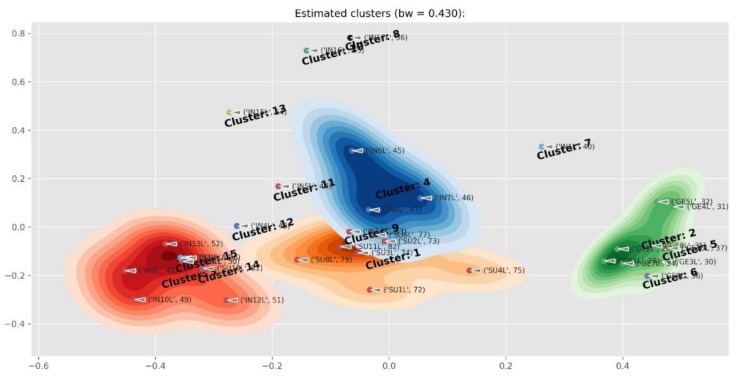
Cluster map of the *Cyclopia intermedia* (IN), *C. genistoides* (GE), and *C. subternata* (SU) leaf extracts with mangiferin and citric acid excluded, showing a separation of *C. genistoides* (green, cluster 2,5, and 6) and from *C. subternata* (orange, cluster 1) and *C. intermedia* (red, cluster 3,4) with some extracts forming additional clusters that appear to be based on geography/provenance/population.

**Figure 8 molecules-24-02352-f008:**
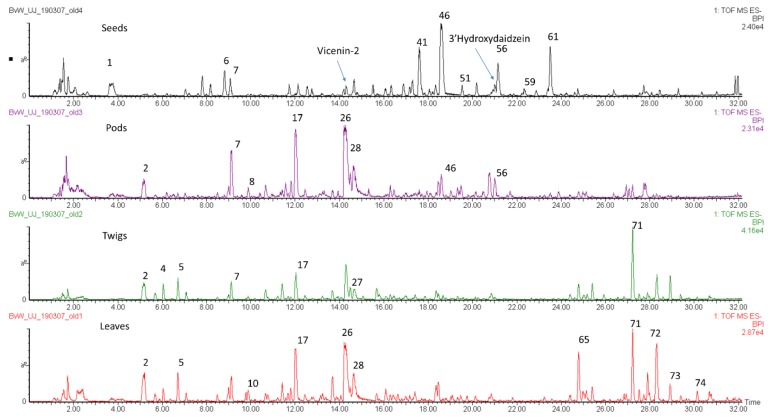
Total ion chromatograms of different plant parts of *Cyclopia aurescens* Kies, showing the seed extract on top with low levels of mangiferin and isomangiferin (compounds **26** and **28**) and large peaks for compounds **46** (naringenin-*O*-hexoside isomer 3), **56** (butein) and **61** (butein-hexoside isomer 2).

**Figure 9 molecules-24-02352-f009:**
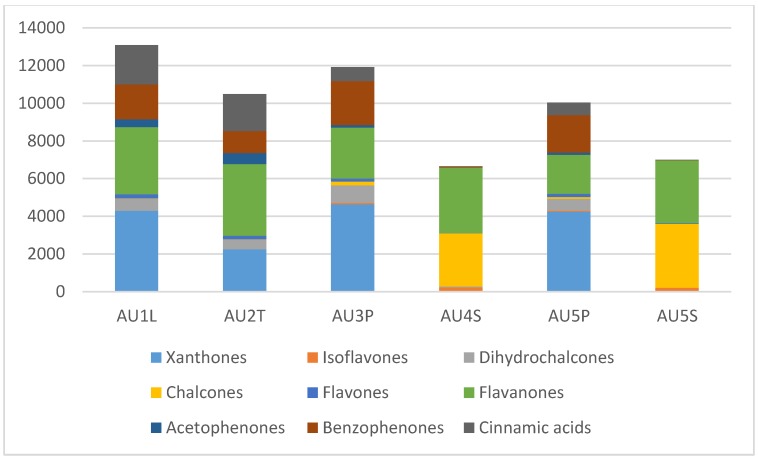
Composition of classes of compounds (as a sum of the concentrations in mg/kg) in various plant parts of *Cyclopia aurescens* (AU1-5, all from Klein Swartberg, refer to Table 2) (L = leaves, T = twigs, P = pods, S = seeds). Leaves, twigs and pods are chemically diverse and have a similar combination of compounds whilst the seeds contain mainly chalcones and flavanones.

**Figure 10 molecules-24-02352-f010:**
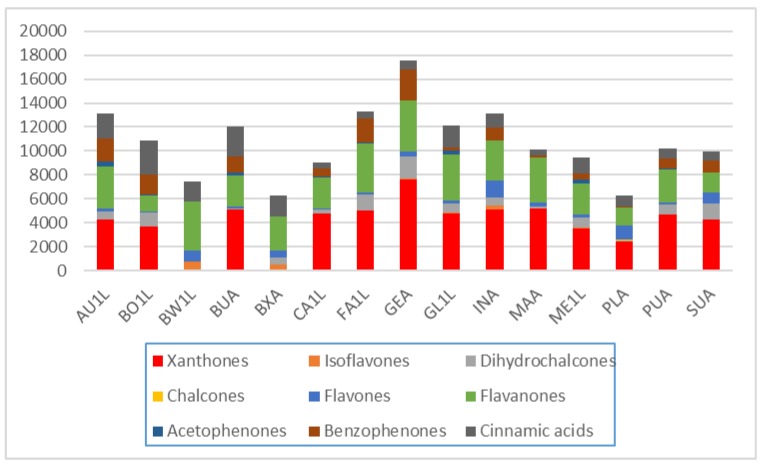
Average levels (mg/kg relative to mangiferin) of nine classes of phenolic compounds in leaf samples of 15 species of *Cyclopia*. The A at the end of the species codes means that it is an average value for all the leaf samples of that species analysed–see Table 2). *C. aurescens* (AU), *C. bolusii* (BO), *C. bowiena* (BW), *C. burtonii* (BU), *C. buxifolia* (BX), *C. capensis* (CA), *C. falcata* (FA), *C. genistoides* (GE), *C. glabra* (GL), *C. intermedia* (IN), *C. maculata* (MA), *C. meyeriana* (ME), *C. plicata* (PL), *C. pubescens* (PU) and *C. subternata* (SU).

**Figure 11 molecules-24-02352-f011:**
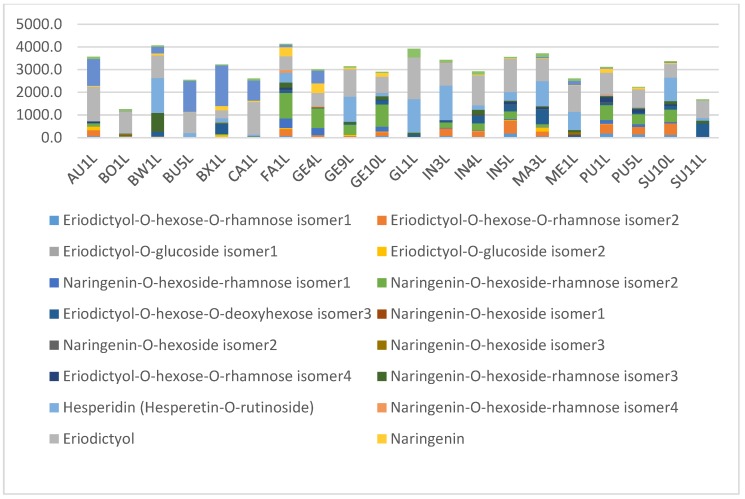
Composition of the flavanones in the leaf (L) samples in the different *Cyclopia* species in mg/kg relative to mangiferin. For sample codes see Table 2). *C. aurescens* (AU), *C. bolusii* (BO), *C. bowiena* (BW), *C. burtonii* (BU), *C. buxifolia* (BX), *C. capensis* (CA), *C. falcata* (FA), *C. genistoides* (GE), *C. glabra* (GL), *C. intermedia* (IN), *C. maculata* (MA), *C. meyeriana* (ME), *C. plicata* (PL), *C. pubescens* (PU) and *C. subternata* (SU).

**Figure 12 molecules-24-02352-f012:**
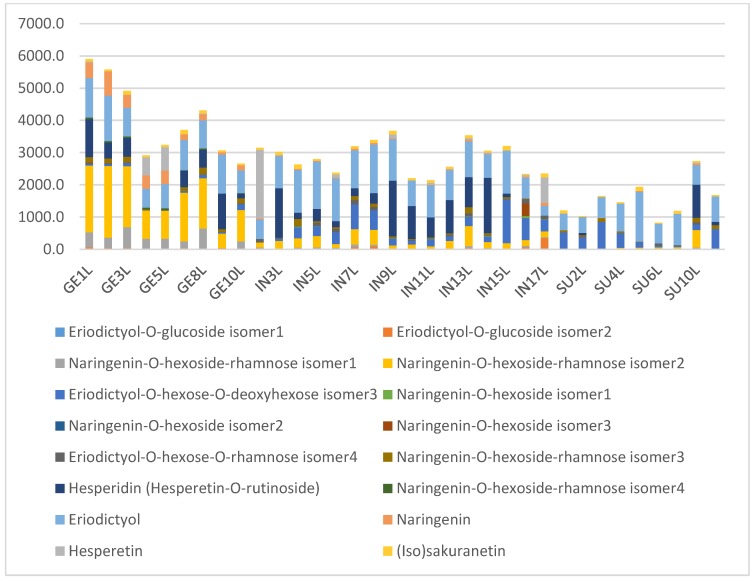
Composition of the flavanones (mg/kg relative to mangiferin) in the leaf samples from the three main commercial sources of honeybush tea: *Cyclopia genistoides* (GE, nine samples), *C. intermedia* (IN, 16 samples) and *C. subternata* (SU, nine samples). For sample codes see Table 2. Numbering is according to the collection point and from West to East in each species.

**Figure 13 molecules-24-02352-f013:**
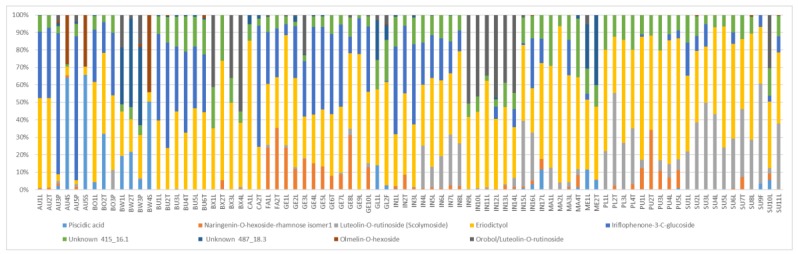
Composition of phenolic compounds relative to the total from the loading plots that caused the separation of clusters (see Figure 7). For sample codes see Table 2, *Cyclopia aurescens* (AU), *C. bolusii* (BO), *C. bowiena* (BW), *C. burtonii* (BU), *C. buxifolia* (BX), *C. capensis* (CA), *C. falcata* (FA), *C. genistoides* (GE), *C. glabra* (GL), *C. intermedia* (IN), *C. maculata* (MA), *C. meyeriana* (ME), *C. plicata* (PL), *C. pubescens* (PU) and *C. subternata* (SU); twigs (T), leaves (L), flowers (F) and pods (P). Numbering is according to the collection point and from West to East in each species.

**Table 1 molecules-24-02352-t001:** List of compounds tentatively identified in *Cyclopia* extracts in this study showing compound number, retention time, detected [M − H] ion, elemental composition and MS^E^ fragments as well as literature references to where the compounds were previously detected.

	Retention Time	Exprimental *m/z*	Formula	MS^E^ Fragments		Reference
**1**	3.64	191.0197	C_6_H_7_O_7_	**191.0197**,111.0087,87.0082,85.0303	*Citric acid	New
**2**	5.18	325.1131	C_12_H_21_O_10_	325.1143,**193.0726**,161.0428,101.0237	Ferulic acid pentoside (arabinose) isomer1	New
**3**	5.73	325.1127	C_12_H_21_O_10_	325.1143,**193.0712**,161.0491,101.0237	Ferulic acid pentoside isomer2	New
**4**	6.06	339.1286	C_13_H_23_O_10_	**339.1292**,193.0725,161.0461,101.0260	Ferulic acid rhamnose isomer1	New
**5**	6.72	339.1286	C_13_H_23_O_10_	339.1268,**207.0880**,178.8859,161.0460,113.0221,101.0234	Ferulic acid rhamnose isomer2	New
**6**	8.82	255.0509	C_11_H_11_O_7_	255.0509,165.0547,72.9930	Piscidic acid	New
**7**	9.13	569.1503	C_25_H_29_O_15_	569.1558,**449.1093****,287.0552**,167.0341,125.0242	Iriflophenone-di-*O,C*-hexoside (3-β-D-glucopyranosyl-4-β-D-glucopyranosyloxyiriflophenone)	[18,19,20,21]
**8**	8.98	153.0189	C_7_H_5_O_4_	**153.0192**,109.0305	*Protocatechuic acid (dihydroxybenzoic acid)	New
**9**	9.79	431.1552	C_19_H_27_O_11_	431.1524,293.0834,233.0672,**89.0247**	Unknown 431_9.8	New
**10**	9.87	423.0922	C_19_H_19_O_11_	423.0932,**303.0525**,193.0142,109.0294	Maclurin-3-*C*-glucoside (3-β-D-glucopyranosylmaclurin)	[18,19,20,21]
**11**	10.77	285.0621	C_12_H_13_O_8_	**285.0622**, 153.0184,152.0117,109.0285,108.0222	Dihydroxybenzoic acid-*O*-pentoside	[19]
**12**	11.8	583.1301	C_25_H_27_O_16_	**583.1302**,421.0778,331.0447,301.0369,272.0332,259.0255	(Iso)Mangiferin-*O*-hexoside (tetrahydroxyxanthone-di-O,C-hexose)	[21]
**13**	11.21	429.1401	C_19_H_25_O_11_	429.1383,**135.0452**	Piceol-hexoside-pentoside isomer1	New
**14**	11.39	429.1400	C_19_H_25_O_11_	429.1404,293.0877,**233.0666,135.0456**	Piceol-hexoside-pentoside isomer2	New
**15**	11.58	431.1555	C_19_H_27_O_11_	**431.1552**,275.0564,163.0406,119.0432	Unknown 431_11.6	New
**16**	11.68	443.1558	C_20_H_27_O_11_	**135.0453**,96.9698	Piceol-hexoside-rhamnoside (Sibiricaphenone)	New
**17**	12.04	407.0981	C_19_H_19_O_10_	407.0979,317.0664,**287.0555**,245.0453,193.0129,125.0247	Iriflophenone-3-*C*-glucoside (3-β-D-glucopyranosyliriflophenone)	[18,19,20,21]
**18**	12.15	417.1046	C_17_H_21_O_12_	**417.1038**,153.0178,152.0110,109.0285,108.0222	Dihydroxybenzoic acid-*O*-dipentoside	[19]
**19**	12.47	325.0918	C_15_H_17_O_8_	325.0942,**163.0406**,119.0503	*p*-Coumaric acid hexoside	New
**20**	12.8	457.1352	C_20_H_25_O_12_	457.1357,**163.0401**,119.0498	*p*-coumaric acid-*O*-pentose-*O*-hexoside1	[19]
**21**	13.15	457.1351	C_20_H_25_O_12_	457.1342,**163.0405**,145.0300,119.0494	*p*-coumaric acid-*O*-pentose-*O*-hexoside2	[19]
**22**	13.27	457.1352	C_20_H_25_O_12_	**457.1342**,163.0403,119.0496	*p*-coumaric acid-*O*-pentose-*O*-hexoside3	[19]
**23**	13.69	401.1446	C_18_H_25_O_10_	**401.1446**,269.1029,179.0345,161.0448,101.0240	Unknown 401_13.6	New
**24**	13.92	595.1644	C_27_H_31_O_15_	**595.1658**,459.1141,433.1251,287.0541,169.0142,161.0269,151.0044,135.0444,125.0245	Eriodictyol-*O*-hexose-*O*-rhamnose isomer1	[18,19,21]
**25**	14.06	457.1709	C_21_H_29_O_11_	457.1703,**293.0873**,233.0671,149.0464,125.0249,89.0246	Unknown 457_14	New
**26**	14.3	421.0764	C_19_H_17_O_11_	**421.0768**,301.0358,331.0441,259.0246	*Mangiferin	[14,17,18,19,21]
**27**	14.48	595.1651	C_27_H_31_O_15_	**595.1658**,459.1141,287.0541,169.0142,161.0269,151.0044,135.0444,125.0245	Eriodictyol-*O*-hexose-*O*-rhamnose isomer2	[18,19,21]
**28**	14.67	421.0763	C_19_H_17_O_11_	**421.0771**,301.0347,331.0458,258.0170	Isomangiferin	[18,19,21]
**29**	14.91	381.1767	C_16_H_29_O_10_	381.1767,249.1344,161.0453,101.0256,**96.9703**	Unknown 381_14.9	New
**30**	15.15	465.1031	C_21_H_21_O_12_	**465.1046**,285.0407,151.0042	Unknown 465_15.15	New
**31**	14.33	449.1079	C_21_H_21_O_11_	449.1081,287.0551,269.0448,259.0616,163.0038,135.0086,121.0290,109.0296	Eriodictyol-*O*-glucoside isomer1	[17]
**32**	15.69	449.1079	C_21_H_21_O_11_	449.1079,287.0553, 269.0450,259.0616,225.0561,**151.0035**,135.0448	Eriodictyol-*O*-glucoside isomer2	[18]
**33**	15.87	579.1725	C_27_H_31_O_14_	**579.1765**,271.0618,151.0027,145.0300,125.0260,119.0489	Naringenin-*O*-hexoside-*O*-rhamnose isomer1	[19]
**34**	16.11	415.1621	C_19_H_27_O_10_	415.1585,273.0681,149.0466,137.0246,101.0249,89.0247	Unknown 415_16.1	New
**35**	16.31	579.1701	C_27_H_31_O_14_	**579.1689**,271.0633,151.0022,145.0282,125.0253,119.0500	Naringenin-*O*-hexoside-*O*-rhamnose isomer2	[19]
**36**	16.47	447.093	C_21_H_19_O_11_	447.0956,**285.0415**,284.0320,255.0299,119.0452,96.9697	Orobol/Luteolin-*O*-hexoside1	New
**37**	16.63	613.1776	C_27_H_33_O_16_	613.1766,505.1346,493.1363,433.1129,403.1020,**373.0938**,331.0838,251.0536,209.0461	3-hydroxyphloretin-3′,5′-di-C-hexoside	[19]
**38**	16.99	463.2177	C_21_H_35_O_11_	**463.2181**,251.0763,191.0575,149.0461,96.9692,89.0249	Unknown 463_17	New
**39**	17.37	463.2186	C_21_H_35_O_11_	**463.2188**,251.0777,191.0567,149.0456,96.9700,89.0250	Unknown 463_17.4	New
**40**	17.55	595.1657	C_27_H_31_O_15_	595.1658,459.1141,433.1251,**287.0541**,169.0142,161.0269,151.0044,135.0444,125.0245	Eriodictyol-*O*-hexose-*O*-rhamnose isomer3	[19]
**41**	17.59	433.1133	C_21_H_21_O_10_	433.1153,**271.0600**,151.0022	Naringenin-*O*-hexoside isomer1	New
**42**	17.88	433.1133	C_21_H_21_O_10_	433.1153,**271.0600**,151.0022	Naringenin-*O*-hexoside isomer2	New
**43**	18.18	593.1505	C_27_H_29_O_15_	**593.1522**,285.0408	*Luteolin-*O*-rutinoside (Scolymoside)	[18,19]
**44**	18.28	487.1812	C_22_H_31_O_12_	**487.1812**,191.0563,149.0456,101.0245,89.0247	Unknown 487_18.3	New
**45**	18.47	597.1815	C_27_H_33_O_15_	597.1801,477.1390,417.1172,387.1068,**357.0969**,209.0449,167.0363, 125.0236	Phloretin-3′,5′-di-*C*-glucoside	[19]
**46**	18.59	433.1129	C_21_H_21_O_10_	433.1133,**271.0607**,135.0452,91.0191	Naringenin-*O*-hexoside isomer3	New
**47**	18.73	447.0942	C_21_H_19_O_11_	**447.0956**,285.0414,284.0334	Orobol/Kaempferol/Luteolin-*O*-hexoside2	New
**48**	19.34	595.1661	C_27_H_31_O_15_	**595.1654**,459.1166,287.0532,161.0247,151.0033,135.0462,125.0247	Eriodictyol-*O*-hexose-*O*-rhamnose isomer4	[19]
**49**	19.53	579.1732	C_27_H_31_O_14_	**579.1657**,271.0623,151.0035,145.0300,125.0260,119.0486,96.9697	Naringenin-*O*-hexoside-*O*-rhamnose isomer3/Narirutin	[21]
**50**	19.83	417.1176	C_21_H_21_O_9_	417.1171,**211.0763**,169.0662,98.0241	Unknown 417_19.8 isomer1	New
**51**	20.17	445.1141	C_22_H_21_O_10_	**445.1138**,283.0615,268.0378,239.0379	Olmelin-*O*-hexoside	New
**52**	20.77	609.1811	C_28_H_33_O_15_	609.1781,**301.0717**,286.0483	*Hesperidin (Hesperetin-*O*-rutinoside)	[18]
**53**	20.74	579.1681	C_27_H_31_O_14_	**579.1765**,271.0617,151.0031,145.0292,125.0250,119.0492,96.9690	Naringenin-*O*-hexoside-*O*-rhamnose isomer4	[19]
**54**	20.99	527.1194	C_26_H_23_O_12_	527.1194, 317.0669, 287.0562,245.0457,193.0141	Unknown 527_20.99	New
**55**	21	593.2447	C_26_H_41_O_15_	547.2388,**515.2121**,96.9693	Unknown 593_21	New
**56**	21.16	271.0612	C_15_H_11_O_5_	271.0612,**135.0449**,96.9697, 91.0187	Butein/Butin	[15]
**57**	21.3	549.1619	C_26_H_29_O_13_	549.1622,301.0710,255.0663,237.0594,**211.0773**,125.0275,89.0239	Unknown 549_21.3	New
**58**	21.67	593.1506	C_27_H_29_O_15_	593.1525,457.1313,417.1021,399.0924,287.0583,163.0395,**152.0112**,**119.0485**,96.9688	Unknown 593_21.6	New
**59**	22.34	433.113	C_21_H_21_O_10_	433.1129,271.0602,**135.0448**,91.0189	Butein-hexoside isomer1	New
**60**	22.3	417.1193	C_21_H_21_O_9_	417.1171,**211.0763**,169.0662,98.0241	Unknown 417_22.3 isomer2	New
**61**	23.49	433.1147	C_21_H_21_O_10_	433.1145,**271.0619**,**135.0456**,91.0194	Butein-hexoside isomer2	New
**62**	24.39	285.0359	C_15_H_9_O_6_	**285.0404**,161.0290,151.0016,135.0422	Orobol	[15,16]
**63**	24.4	287.0561	C_15_H_11_O_6_	287.0561,151.0038,**135.0452**	Eriodictyol	[26]
**64**	24.7	593.1856	C_28_H_33_O_14_	593.1882,**285.0759**,243.0666,151.0045	Didymin/Neoponcirin (Isosakuranetin-7-*O*-rutinoside)	New
**65**	24.79	447.2226	C_21_H_35_O_10_	447.2246,315.1848,**161.0459**,101.0243,96.9688,113.0239,71.0130	Unknown 447_25	New
**66**	25.03	285.0402	C_15_H_9_O_6_	**285.0404**,175.0396,151.0051,133.0301	*Luteolin	[25]
**67**	25.35	285.0783	C_16_H_13_O_5_	**285.0400**,255.0698,163.0379,135.0315	Unknown 285_25.35	New
**68**	25.93	301.2021	C_16_H_29_O_5_	**301.2024**,96.9695	Unknown 301_25.9	New
**69**	26.36	271.0609	C_15_H_11_O_5_	271.0620,151.0036,**119.0500**,107.0136,96.9683	*Naringenin	[26]
**70**	27.05	327.217	C_18_H_31_O_5_	**327.2184**,229.1416,211.1331,171.1022	Unknown 327_27	New
**71**	27.23	301.0713	C_16_H_13_O_6_	**301.0712**,286.0497,164.0111,151.0034,136.0181	*Hesperetin	[15,26]
**72**	28.29	287.2221	C_16_H_31_O_4_	**287.2211**,96.9678,78.9490	Unknown 287_28.3 (hydroxylated fatty acid?)	New
**73**	28.92	285.2067	C_16_H_29_O_4_	**285.2070**,96.9668	Unknown 285_28.92	New
**74**	30.15	285.0763	C_16_H_13_O_5_	**285.0760**,270.0516,243.0666,164.0114,151.0030,136.0164,108.0216	(Iso)sakuranetin	[14]

*Standard was use to confirm retention time and spectra, base peaks in MS^E^ fragmentation data in bold.

**Table 2 molecules-24-02352-t002:** A list of the samples, their species, sample codes, voucher numbers, collection dates and localities, numbered from West to East per species.

Sample Number	Species	Sample Code	Provenance	VOUCHER SPECIMEN	Date	Part(s) Analysed
					Collected	
1	***Cyclopia aurescens* Kies**	**AU1L**	Klein Swartberg	*Schutte & Van Wyk 771a*	3/2/1992	leaves
2		AU2T	Klein Swartberg	*Schutte & Van Wyk 771a*	3/2/1992	twigs
3		AU3P	Klein Swartberg	*Schutte & Van Wyk 771a*	3/2/1992	pods
4		AU4S	Klein Swartberg	*Schutte & Van Wyk 771a*	3/2/1992	seeds
5		AU5P	Klein Swartberg	*Schutte & Van Wyk 775*	3/2/1992	pods
6		AU5S	Klein Swartberg	*Schutte & Van Wyk 775*	3/2/1992	seeds
7	***Cyclopia bolusii* Hofmeyr & E.Phillips**	**BO1L**	Swartberg Pass	*Schutte & Vlok 749*	1/2/1992	leaves
8		BO2T	Swartberg Pass	*Schutte & Vlok 749*	1/2/1992	twigs
9		BO3P	Swartberg Pass	*Schutte & Vlok 749*	1/2/1992	pods
10	***Cyclopia bowieana* Harv.**	**BW1L**	Ruitersberg	*Schutte 526*	1/1990	leaves
11		BW2T	Ruitersberg	*Schutte 526*	1/1990	twigs
12		BW3P	Ruitersberg	*Schutte 526*	1/1990	pods
13		BW4S	Ruitersberg	*Schutte 526*	1/1990	seeds
14	***Cyclopia burtonii* Hofmeyr & E.Phillips**	**BU1L**	Swartberg	*Schutte 641*	9/1990	leaves
15		BU2T	Swartberg	*Schutte 641*	9/1990	twigs
16		**BU3L**	Swartberg Pass	*Schutte 643*	9/1990	leaves
17		BU4T	Swartberg Pass	*Schutte 643*	9/1990	twigs
18		**BU5L**	Swartberg Pass	*Schutte 747*	1/2/1992	leaves
19		BU6T	Swartberg Pass	*Schutte 747*	1/2/1992	twigs
20	***Cyclopia buxifolia* (Burm.f.) Kies**	**BX1L**	Jonkershoek	*Schutte 604*	9/1990	leaves
21		BX2T	Jonkershoek	*Schutte 604*	9/1990	twigs
22		**BX3L**	Jonkershoek	*Schutte 605*	9/1990	leaves
23		**BX4L**	Jonkershoek	*Schutte 606*	9/1990	leaves
24	***Cyclopia capensis* T.M.Salter**	**CA1L**	Cape Point	*Schutte 550*	1/1990	Leaves
25		CA2T	Cape Point	*Schutte 550*	1/1990	twigs
26	***Cyclopia falcata* (Harv.) Kies**	**FA1L**	Franschoek Pass	*Schutte 612*	9/1990	leaves
27		FA2T	Franschhoek Pass	*Schutte 612*	9/1990	twigs
28	***Cyclopia genistoides* (L.) R.Br.**	**GE1L**	Constantia Mountain	*Schutte 615*	14/09/1990	leaves
29		**GE2L**	Constantia Mountain	*Van Wyk 2747*	16/1/1988	leaves
30		**GE3L**	Rooiels	*Schutte 622*	15/9/1990	leaves
31		**GE4L**	Bettys Bay	*Schutte 624*	15/9/1990	leaves
32		**GE5L**	Bettys Bay	*Schutte 624*	15/9/1990	leaves
33		GE6T	Bettys Bay	*Schutte 624*	15/9/1990	twigs
34		**GE7L**	Bettys Bay	*Schutte 624*	15/9/1990	leaves
35		**GE8L**	Bettys Bay	*Schutte 625*	15/9/1990	leaves
36		**GE9L**	Buffelshoek, Albertinia	*Vlok 2249*	28/11/1989	leaves
37		**GE10L**	De Hoop	*Boatwright & Magee 53*	25/11/2004	leaves
38	***Cyclopia glabra* (Hofmeyr & E.Phillips) A.L. Schutte**	**GL1L**	Matroosberg	*Schutte 557*	01/2/1990	leaves
39		GL2F	Matroosberg	*Schutte 557*	01/2/1990	flowers
40		**IN1L**	Anysberg	*Schutte 680*	9/1990	leaves
41	***Cyclopia intermedia* E.Mey.**	IN2T	Anysberg	*Schutte 680*	9/1990	twigs
42		**IN3L**	Touwsberg	*Van Wyk, Winter & Tilney 3416*	05/10/1993	leaves
43		**IN4L**	Oudtshoorn	*Schutte 521*	24/1/1990	leaves
44		**IN5L**	Teeberg	*Schutte 524*	25/1/1990	leaves
45		**IN6L**	Teeberg	*Schutte 724b & c*	1/1992	leaves
46		**IN7L**	Swartberg Pass	*Schutte 646*	17/9/1990	leaves
47		**IN8L**	Swartberg Pass	*Schutte 647*	17/9/1990	leaves
48		**IN9L**	Prince Alfred’s Pass	*Van Wyk 928*	20/2/1982	leaves
49		**IN10L**	Prince Alfred’s Pass	*Schutte 578*	23/1/1990	leaves
50		**IN11L**	K’Buku, De Vlug	*Van Wyk 945*	20/2/1982	leaves
51		**IN12L**	K’Buku, De Vlug	*Van Wyk 947*	20/2/1982	leaves
52		**IN13L**	K’Buku, De Vlug	*Van Wyk 951*	20/2/1982	leaves
53		**IN14L**	Joubertina	*Schutte 507*	22/01/1990	leaves
54		**IN15L**	Hoopsberg	*Schutte 513*	23/01/1990	leaves
55		**IN16L**	Hoopsberg	*Schutte 573*	1/1990	leaves
56		**IN17L**	Hoopsberg	*Schutte 573*	1/1990	twigs
57	***Cyclopia maculata* (Andrews) Kies**	**MA1L**	Garcia State Forest	*Schutte 528b*	26/01/1990	leaves
58		**MA2L**	Garcia State Forest	*Van Wyk 895*	02/10/1981	leaves
59		**MA3L**	Garcia State Forest	*Schutte 528*	1/1990	leaves
60		MA4T	Garcia State Forest	*Schutte 528*	1/1990	twigs
61	***Cyclopia meyeriana* Walp.**	**ME1L**	Matroosberg	*Schutte 557*	1/2/1990	leaves
62		ME2T	Matroosberg	*Schutte 557*	1/2/1990	twigs
63	***Cyclopia plicata* Kies**	**PL1L**	Hoopsberg	*Schutte 670a*	09/1990	leaves
64		PL2T	Hoopsberg	*Schutte 670a*	09/1990	twigs
65		**PL3L**	Hoopsberg	*Schutte 670b*	09/1990	leaves
66		PL4T	Hoopsberg	*Schutte 670b*	09/1990	twigs
67	***Cyclopia pubescens* Eckl. & Zeyh.**	**PU1L**	Port Elizabeth	*Schutte 685*	22/9/1990	leaves
68		PU2T	Port Elizabeth	*Schutte 685*	22/9/1990	twigs
69		**PU3L**	Port Elizabeth	*Schutte 686*	22/9/1990	leaves
70		**PU4L**	Port Elizabeth	*Schutte 687*	22/9/1990	leaves
71		**PU5L**	Port Elizabeth	*Schutte 688*	22/9/1990	leaves
72	***Cyclopia subternata* Vogel**	**SU1L**	Bloukrantz River	*Schutte 683*	21/09/1990	leaves
73		**SU2L**	Bloukrantz River	*Schutte 683*	9/1990	leaves
74		**SU3L**	Kareedouw Pass	*Schutte 505*	22/01/1990	leaves
75		**SU4L**	Prince Alfred’s Pass	*Schutte 519*	23/01/1990	leaves
76		**SU5L**	Prince Alfred’s Pass	*Van Wyk 939*	20/2/1985	leaves
77		**SU6L**	Outeniqua Pass	*Schutte 639*	9/1990	leaves
78		SU7T	Outeniqua Pass	*Schutte 639*	9/1990	twigs
79		**SU8L**	Outeniqua Pass	*Schutte 690b*	08/09/1991	leaves
80		SU9F	Outeniqua Pass	*Schutte 690b*	08/09/1991	flowers
81		**SU10L**	Witelsbos	*Schutte 503*	22/01/1990	leaves
82		**SU11L**	Elandsbos River	*Schutte s.n. 1b*	9/1990	leaves

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
