# Peer review of "Patterns of Variation and Chemosystematic Significance of Phenolic Compounds in the Genus Cyclopia (Fabaceae, Podalyrieae)"

_molecules, 2019, doi:10.3390/molecules24132352_

Round 1

Reviewer 1 Report

In the present study, 82 samples from 15 of the 23 known Cyclopia species were analyzed by using liquid-chromatography/high resolution mass spectrometry (LC/HRMS). In addition, principle components analysis for these data of chemical compositions were performed and the results indicated that a combination of compounds have potential as diagnostic characters at species level. Although I appreciated authors′ efforts, the present research work only displayed preliminary advance and exhibited several technical flaws. Therefore, this manuscript is not recommended to accept for publication in Molecules in the present form. It should be reconsidered the acceptance after authors′ revisions. In addition, there were some major comments addressed as following.

1.         There were various typographic, grammar, and format errors to be found in the text. Authors have to check and revise these errors following the guidelines of this journal.

2.         Table 1 and Figure 2 should move to Supplementary Materials.

3.         Line 76, the first appeared compound in the article is 12. It is not the common way. Authors have to number the compound according to the appearance sequence.

4.         In the tentative identification section, there were many compounds identified as hexoside or pentoside. These were not real structural identification and these results were not helpful for the chemistry researchers. In addition, how could authors confirm that these isomers were always the same one in different species?

5.         Lines 117-121, these results demonstrated that the concentration reported in the present study was not reliable. What are these results used for? Line 121, authors have to note that these data were provided in the Supplementary Materials.

6.         Line 167, authors have to note that data were not shown.

7.         Figure 12, several tentatively identified isomers were quantified and compared. How could authors confirm that these isomers were always the same one in different species? If they are different compounds, how did authors quantify them?

8.         Lines 210-211 (Figure 13), authors declared that eight compounds should be identified with high priority. However, two were unknown compounds. How to “identify” unknown compounds to diagnose the chemical identities?

9.         The last sentence in Conclusion section was overclaimed according to the present results.

10.     The Materials and Methods section was too brief. Authors have to rewrite this section according to the common styles appeared in other articles.

11.     In the References section, the writing manner of references did not follow the style of this journal strictly. Authors have to check and revise the style and typographic errors in this section carefully.

Author Response

1.           There were various typographic, grammar, and format errors to be found in the text. Authors have to check and revise these errors following the guidelines of this journal.

The manuscript was reviewed and the grammar and format errors corrected.

2.         Table 1 and Figure 2 should move to Supplementary Materials.

Table 1 was moved to the Materials and methods section as was requested by the other reviewer

3.         Line 76, the first appeared compound in the article is 12. It is not the common way. Authors have to number the compound according to the appearance sequence.

The compounds are labelled in order of their retention times.  The numbering was not changed, but we have moved the table above the text so that it is referred first and in numerical order.

4.         In the tentative identification section, there were many compounds identified as hexoside or pentoside. These were not real structural identification and these results were not helpful for the chemistry researchers. In addition, how could authors confirm that these isomers were always the same one in different species?

The type of pentoside or hexoside can only be elucidated if NMR was used (or if both occur in the samples, galactosides eg always elute before glucosides), or if an authentic standard was available. These compounds are easily separated using this chromatographic system.  The compounds between species had the same retention times, accurate mass and fragmentation spectra.

Refer to Abad-Garcia et al, 2009, J. Chrom A, 1216, 5398-5415.

5.         Lines 117-121, these results demonstrated that the concentration reported in the present study was not reliable. What are these results used for? Line 121, authors have to note that these data were provided in the Supplementary Materials.

The mangiferin concentrations were very high for MS and were quantified in addition using UV.  This value was left out in some of the PCAs because it skewed the results.  The reference to the supplementary material was added.

6.         Line 167, authors have to note that data were not shown.

This was noted.

7.         Figure 12, several tentatively identified isomers were quantified and compared. How could authors confirm that these isomers were always the same one in different species? If they are different compounds, how did authors quantify them?

The retention times and fragmentation data were the same.  The quantifications were all done relative to a mangiferin standard.  There are no commercially available standards for all these compounds. Results are seen as relative to the mangiferin standard.

8.         Lines 210-211 (Figure 13), authors declared that eight compounds should be identified with high priority. However, two were unknown compounds. How to “identify” unknown compounds to diagnose the chemical identities?

The unknown compounds should be isolated and structure elucidation performed using NMR. “This would require isolation and purifying the compounds and confirmation and structural elucidation using Nuclear Magnetic Resonance spectroscopy (NMR). “ was added to the text to clarify.

9.         The last sentence in Conclusion section was overclaimed according to the present results.

We do not agree with the statement, since the last sentence states something that is a possibility for the future and not limited to the current study.

10.     The Materials and Methods section was too brief. Authors have to rewrite this section according to the common styles appeared in other articles.

We have reviewed the materials and methods and made sure that it contained all the experimental conditions. 

11.     In the References section, the writing manner of references did not follow the style of this journal strictly. Authors have to check and revise the style and typographic errors in this section carefully.

References were edited and formatted.

Reviewer 2 Report

General comments:

Although this is an interesting study, authors present so many irrelevant information that is a distraction from the main goal of the research. For instance, there is a table mixing the results from the present research with those found in literature, however, there is not an explanation and discussion of their own results. In addition, a lot number of tables and figures are not well explained in the discussion of the manuscript.

In my opinion, authors need to highlight the importance of their results and discuss more in deep their own results.

Abstract

Lines 16: Please, write the sentence in impersonal form (instead of “we analysed”, “we decided to…”) as required in scientific papers. Correct lines 58-59 “we decided to…”; line 130 “we have confirmed”; line 133 “We investigated…”; line 135 “on our…”; line 139 “We did not detect…”

Lines 20-21: Authors should provide more specific results in “some consistent differences between species and provenances were observed”

In my opinion, results presented in the abstract are ambiguous. Authors should provide information in basis on the aim of the research “The aim was to determine if different species and populations of Cyclopia can be distinguished by quantitative and perhaps also qualitative differences in their overall phenolic profiles”.

Results and discussion

Lines 67-68: “Table 1 contains a list of the samples and their species, voucher numbers and the localities where they were collected”. Please, change this information, as well as table 1, to the material and methods section.

Line 68: Table 2 contain results from other researchers. It is strange to include this information in the section were authors should show their own results. Authors should specify that the information comes from other studies, as well as clarifying which phenolic compounds they detected in this research.

Lines 70-72. Are authors referring to Table 2? Please, specify. Line 71: There are more references contained in table 2. Please, add the missing information.

Lines 67-72. Discuss all these results and clarify the main points of this information.

Line 75: Which Table or Figure contain the information of this section? Please, specify.

Line 107. Are authors shure that mangiferin (compound 26) in the top two plants from Figure 1 is absent? There is a small peak in both chromatogram (22-BX4L and 13-BW4S) which could correspond to peak 26. Maybe mangiferin is out of the detection limit.  Authors must add the name of each compound detected in the peaks.

Line 131: Please, specify where the information is shown (Figure or Table number).

Line 151: Were the chemical structures obtained from this research? In my opinion, this information is irrelevant.

Figures 1– 7: There are not explained the information of all these figures in the section of results and discussion.

Line 169-182. The result must be presented as “concentration of each phenolic compound” independently of the variety or part of analyzed plant. Authors only present a large number of figures which are not well explained and lacking of relevant information: Figure 1 with 5 chromatograms: Figure 3 with 2 chromatograms, Figure 8 with 4 chromatograms. It could be more interesting if adding table(s) with the “concentration of phenolic compounds” instead of adding a large number of Figures (chromatograms).

Lines 184-187. Figure 8: Missing information regarding the name of each compounds in the corresponding peaks.

Lines 189-191. Figure 9. Missing information regarding the meaning of AU1L, AU2T, AU3P, AU4S, AU5P and AU5S. What are the units of numbers in axis “y”?

Line 193-194. “The results suggest that phenolic compounds do have diagnostic value to distinguish between some of the species, especially when used in combination” for me is not clear this statement mainly taking into account the obtained results…. As stated in line 200 “no clear pattern and the underlying processes (phenotypic or genetic) deserve more detailed studies” this statement is true.

Lines 232-234. Figure 10. Missing information regarding the meaning of AU1L, BO1L, BW1L… etc.  What are the units of numbers in axis “y”?

Line 236-237. Figure 11. What are the units of numbers in axis “y”?

Lines 239-241. Figure 12. What are the units of numbers in axis “y”?

Conclusion

Line 254. The HPLC analysis was not part of the methodology of the present research. Authors used a “High resolution UPLC-MS analysis” as stated in line 287.  So, the statement “The study also revealed that the methods of extraction and analysis by HPLC influence the results” was not obtained from your own research. Please, change the sentence.

Author Response

General comments:

Although this is an interesting study, authors present so many irrelevant information that is a distraction from the main goal of the research. For instance, there is a table mixing the results from the present research with those found in literature, however, there is not an explanation and discussion of their own results. In addition, a lot number of tables and figures are not well explained in the discussion of the manuscript.

In my opinion, authors need to highlight the importance of their results and discuss more in deep their own results.

Table 2 contains all the compounds detected in this study, the reference column shows in which papers it was also previously detected (in some cases other techniques including NMR was used). We have added a sentence to make it clearer and have also added new to all the compounds that were detected for the first time in Cyclopia.

Abstract

Lines 16: Please, write the sentence in impersonal form (instead of “we analysed”, “we decided to…”) as required in scientific papers. Correct lines 58-59 “we decided to…”; line 130 “we have confirmed”; line 133 “We investigated…”; line 135 “on our…”; line 139 “We did not detect…”

The impersonal form was removed from the text.

Lines 20-21: Authors should provide more specific results in “some consistent differences between species and provenances were observed”

Noted, we have rewritten parts of the text in the discussion and have made added more information to the figures to highlight the results better.

In my opinion, results presented in the abstract are ambiguous. Authors should provide information in basis on the aim of the research “The aim was to determine if different species and populations of Cyclopia can be distinguished by quantitative and perhaps also qualitative differences in their overall phenolic profiles”.

“ 74 Phenolic compounds are presented, many of which were identified for the first time in Cyclopia species, with 11 of these being responsible for the separation between samples in the PCAs.” Was added to the abstract

Results and discussion

Lines 67-68: “Table 1 contains a list of the samples and their species, voucher numbers and the localities where they were collected”. Please, change this information, as well as table 1, to the material and methods section.

Table 1 was moved to Material and methods and its heading changed.

Line 68: Table 2 contain results from other researchers. It is strange to include this information in the section were authors should show their own results. Authors should specify that the information comes from other studies, as well as clarifying which phenolic compounds they detected in this research.

This table contains the results from this study, we have changed the heading to make it clearer, the references are to show where are people have also detected the same compounds, many of these references used other techniques like NMR.

We have edited the text as well.

Lines 70-72. Are authors referring to Table 2? Please, specify. Line 71: There are more references contained in table 2. Please, add the missing information.

This was corrected and the references added.

Lines 67-72. Discuss all these results and clarify the main points of this information.

The results and discussion was edited and the descriptions of figures edited to clarify.

Line 75: Which Table or Figure contain the information of this section? Please, specify.

The table name was added to the heading.

Line 107. Are authors sure that mangiferin (compound 26) in the top two plants from Figure 1 is absent? There is a small peak in both chromatogram (22-BX4L and 13-BW4S) which could correspond to peak 26. Maybe mangiferin is out of the detection limit.  Authors must add the name of each compound detected in the peaks.

All compounds detected are in Table 1 and Table Sup 1, the peak the referee is referring to in sample (22-BX4L and 13-BW4S) is not mangiferin (See Figure 8, it is vicenin 2 in the seed sample, 13BW4S). There also another peak eluting between the two, namely Eriodictyol-O-hexose-O-rhamnose isomer2, compound 27, which is the peak in BX4L.

Line 131: Please, specify where the information is shown (Figure or Table number).

Table 1, it was added.

Line 151: Were the chemical structures obtained from this research? In my opinion, this information is irrelevant.

Yes, this is the most comprehensive study to date, many of the compounds are reported for the first time.

Figures 1– 7: There are not explained the information of all these figures in the section of results and discussion.

A paragraph was added to explain these figures better.

Line 169-182. The result must be presented as “concentration of each phenolic compound” independently of the variety or part of analyzed plant. Authors only present a large number of figures which are not well explained and lacking of relevant information: Figure 1 with 5 chromatograms: Figure 3 with 2 chromatograms, Figure 8 with 4 chromatograms. It could be more interesting if adding table(s) with the “concentration of phenolic compounds” instead of adding a large number of Figures (chromatograms).

The concentrations of the Phenolic compounds are in Supplementary table 1 and are summarized in figure 2.  The figures were edited to make the text more descriptive.  Many of the figures already contained concentrations, the text and descriptions were edited to clarify.

Lines 184-187. Figure 8: Missing information regarding the name of each compounds in the corresponding peaks.

There are 74 compounds that were detected in the samples which makes it not possible to label all peaks, Table 1 contains the retention times as reference.  We labelled more peaks to clarify it more.

Lines 189-191. Figure 9. Missing information regarding the meaning of AU1L, AU2T, AU3P, AU4S, AU5P and AU5S. What are the units of numbers in axis “y”?

Mg/kg, we have added it to the text, more information on the codes were added

Line 193-194. “The results suggest that phenolic compounds do have diagnostic value to distinguish between some of the species, especially when used in combination” for me is not clear this statement mainly taking into account the obtained results…. As stated in line 200 “no clear pattern and the underlying processes (phenotypic or genetic) deserve more detailed studies” this statement is true.

This paragraph was edited to make it clearer.

Lines 232-234. Figure 10. Missing information regarding the meaning of AU1L, BO1L, BW1L… etc.  What are the units of numbers in axis “y”?

Mg/kg, we have added it to the text, more information on the codes were added

Line 236-237. Figure 11. What are the units of numbers in axis “y”?

Mg/kg, we have added it to the text

Lines 239-241. Figure 12. What are the units of numbers in axis “y”?

Mg/kg, we have added it to the text

Conclusion

Line 254. The HPLC analysis was not part of the methodology of the present research. Authors used a “High resolution UPLC-MS analysis” as stated in line 287.  So, the statement “The study also revealed that the methods of extraction and analysis by HPLC influence the results” was not obtained from your own research. Please, change the sentence.

HPLC was replaced by UPLC-HRMS analysis.

Round 2

Reviewer 1 Report

The authors have tried hard to answer all my questions adequately. Therefore, the revised manuscript is recommended for publication. 

Author Response

Thank you for your review!

Reviewer 2 Report

Authors have now improved the manuscript.

There is a small mistake in the document that authors should correct (table numeration):

Line 329: This is Table 2. Please, revise all the document taking into account this point (i.e. line 334-335).

Author Response

Thank you the change has been made.